# Intraductal Papillary Mucinous Neoplasms and GLP-1 Receptor Agonists: Navigating Therapeutic Uncertainty in Diabetes Management

**DOI:** 10.3390/biomedicines13102326

**Published:** 2025-09-23

**Authors:** Francesco Tassone, Giovanna Saraceno

**Affiliations:** Territorial Diabetes Care Unit, ASL TO5, 10023 Chieri, Italy

**Keywords:** pancreatic cystic neoplasms, incretin therapy, diabetes complications, pancreatic cancer, risk stratification

## Abstract

The management of type 2 diabetes in patients with intraductal papillary mucinous neoplasms (IPMNs) presents a growing clinical dilemma. As glucagon-like peptide-1 receptor agonists (GLP-1RAs) become first-line therapies for diabetes and obesity, their safety in patients with premalignant pancreatic lesions remains uncertain. This viewpoint examines current evidence through three critical lenses: molecular mechanisms linking incretin signaling to IPMN progression, clinical outcomes from large-scale pharmacovigilance studies, and practical management considerations. We propose a risk-stratified approach that balances the proven metabolic benefits of GLP-1RAs against theoretical oncogenic risks, emphasizing the need for shared decision-making and enhanced surveillance protocols.

## 1. Introduction

The last decade has witnessed the convergence of two major phenomena in clinical practice. On the one hand, advances in imaging technology have resulted in an unprecedented detection rate of incidental pancreatic cystic lesions, many of which are intraductal papillary mucinous neoplasms (IPMNs). On the other hand, the management of type 2 diabetes has been revolutionized by the introduction of glucagon-like peptide-1 receptor agonists (GLP-1RAs), which are now considered cornerstone therapies due to their remarkable efficacy in glycemic control, weight reduction, and cardiovascular protection [1,2,3].

This intersection generates a therapeutic dilemma: how should clinicians navigate diabetes management in patients with co-existing IPMNs? Should GLP-1RA therapy be withheld due to theoretical risks of neoplastic progression or embraced for its undeniable systemic benefits? The lack of randomized trials addressing this specific population leaves uncertainty.

As practicing clinicians, we now encounter these patients more frequently than ever before. Our task is not only to summarize available data but to interpret them, share the discomfort we face when evidence is incomplete, and propose a pragmatic framework for decision-making. In this viewpoint essay, we argue that while theoretical concerns cannot be ignored, most patients with low-risk IPMNs should not be deprived of GLP-1RA therapy. The real challenge lies in balancing scientific evidence, biological plausibility, and the lived realities of patient care.

## 2. IPMN Biology and Risk Stratification

IPMNs are mucin-producing epithelial neoplasms of the pancreatic ducts and recognized precursors of pancreatic ductal adenocarcinoma. They are classified into branch-duct (BD-IPMN), main-duct (MD-IPMN), or mixed-type adenocarcinomas, each carrying a different degree of malignant potential [4,5].

Recent updates to the Kyoto 2024 guidelines [6] highlight that although radiological criteria remain the cornerstone of risk stratification, they are far from perfect. High-risk stigmata (such as main duct dilation ≥ 10 mm or the presence of enhancing mural nodules) warrant surgical consideration, while “worrisome features” (such as a cyst size ≥ 3 cm, thickened walls, or elevated serum CA 19-9 levels) suggest a need for closer surveillance. Still, many lesions fall into gray zones.

Molecular analyses have revealed driver mutations such as KRAS and GNAS that precede malignant transformation [7]. However, integration of molecular biomarkers into routine care remains aspirational rather than operational. For now, surveillance relies primarily on cross-sectional imaging, often leaving clinicians with the uneasy feeling of waiting while uncertainty lingers.

The implication for diabetes management is direct: if GLP-1RAs were to accelerate neoplastic transformation, even a small incremental risk could have profound implications for these already high-risk patients. Conversely, overestimating this risk may lead to withholding therapies that confer significant cardiovascular protection.

## 3. GLP-1 Receptor Agonists: Mechanism of Action and Pancreatic Effects

GLP-1RAs have transformed type 2 diabetes management. Beyond lowering HbA1c by up to 1.5%, they induce 5–15% weight loss and significantly reduce major adverse cardiovascular events [2,3,8]. For patients with obesity and cardiometabolic risk, they are arguably the most impactful therapy currently available.

The biological plausibility of pancreatic harm stems from early preclinical studies indicating ductal proliferation with GLP-1 stimulation [9]. However, these findings were inconsistent and largely confined to rodent models. Subsequent studies on nonhuman primates and humans have not demonstrated the same proliferative effects [10].

Still, uncertainty persists. Case reports and pharmacovigilance signals initially raised concerns of pancreatitis and pancreatic cancer [11]. Large randomized controlled trials such as LEADER (liraglutide) and SUSTAIN (semaglutide) were not designed to assess neoplastic outcomes, though they reported no excess in pancreatic events [12].

As clinicians, our unease lies not in proven harm but in the absence of definitive reassurance. For a patient with an incidental IPMN, the resulting question is as follows: should we rely on indirect evidence or defer therapy until more is known?

## 4. Clinical Evidence: Reconciling Theory and Reality

Recent years have brought stronger reassurance.

A Danish nationwide registry study (2025) including >145,000 GLP-1RA users found no increased long-term risk of pancreatic cancer compared to other antidiabetic drugs [13].

A population-based cohort analysis (2025) reported no excess incidence of pancreatitis or pancreatic cancer following GLP-1RA initiation [14].

A 2025 meta-analysis of randomized controlled trials confirmed there was no significant increase in overall cancer risk among GLP-1RA users [15].

Updated gastroenterology reviews emphasize that while vigilance is warranted, there is no convincing evidence on whose basis GLP-1RA use can be contraindicated for patients with incidental pancreatic cysts [5,6].

These findings should reassure us, yet they do not directly answer the specific question regarding IPMN-bearing patients. No prospective studies have stratified outcomes by cystic lesion subtype; thus, the absence of harm is inferred rather than proven.

From our perspective, this is precisely where an opinion piece must transcend data—acknowledging the gap, articulating the unease, and proposing a way forward.

## 5. A Pragmatic Management Framework

Our approach is pragmatic, rooted in patient-centered decision-making and tempered by caution. We propose the following framework (summarized in Table 1):Low-risk BD-IPMNs (<3 cm, no worrisome features): GLP-1RA therapy may be initiated, provided that routine imaging surveillance is maintained. The systemic benefits outweigh the theoretical risks.High-risk or MD-IPMNs (a main duct ≥ 10 mm, mural nodules, or high-risk stigmata): Defer GLP-1RA therapy until multidisciplinary evaluation is complete. In the meantime, consider alternative glucose-lowering agents with a neutral pancreatic profile (e.g., SGLT2 inhibitors and metformin).

Clinical Scenario 1

A 58-year-old woman with obesity, uncontrolled type 2 diabetes (HbA1c 9.2%), and a 1.8 cm BD-IPMN without worrisome features is eager to begin semaglutide for weight loss. In this case, we would initiate therapy, ensuring annual MRI surveillance and detailed counseling about warning signs. The potential cardiometabolic benefit is substantial, and the risk remains theoretical.

Clinical Scenario 2

A 66-year-old man with coronary artery disease, insulin-treated diabetes, and an MD-IPMN with duct dilation of 12 mm is being followed by a pancreatic surgical team. Here, GLP-1RA therapy should be deferred, as the risk of malignancy is already high and the theoretical contribution of GLP-1 signaling cannot be ignored. Alternative agents should be prioritized until a definitive surgical decision is made.

These examples reflect the reality of clinical practice: not a binary choice but a calibrated balance between risks, benefits, and patient values.

**Table 1 biomedicines-13-02326-t001:** Proposed management framework for GLP-1RA use for patients with IPMNs.

Patient with Diabetes and IPMN.	
**Branch-Duct IPMN** <3 cm, no worrisome features	**Main-Duct IPMN or High-Risk Stigmata** (duct ≥ 10 mm, mural nodules, etc.)
→ GLP-1RA appropriate (with annual imaging surveillance)	→ Defer GLP-1RA Consider alternatives (SGLT2 inhibitors and metformin) Multidisciplinary evaluation

## 6. Conclusions and Future Directions

In our view, the fear of GLP-1RA-induced IPMN progression is disproportionate to the evidence available. For the majority of patients with low-risk IPMNs, denying GLP-1RA therapy risks depriving them of meaningful metabolic and cardiovascular benefits. Yet we must candidly acknowledge that uncertainty persists, and that clinicians are justified in their discomfort.

Looking forward, the path to clarity requires the following:

Prospective registries of patients with IPMNs exposed to incretin therapies;Smarter imaging tools capable of detecting subtle progression earlier;Integration of molecular biomarkers into surveillance protocols;Artificial-intelligence-driven risk models that combine imaging, molecular, and clinical data to personalize therapy.

This is not the end of the discussion but the beginning. As clinicians, we must confront uncertainty not by avoiding difficult choices but by engaging with them transparently. Our hope is that by articulating these dilemmas, sharing our perspectives, and proposing pragmatic frameworks, we can contribute to collective progress.

## Data Availability

No new data were created.

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
