# Peer review of "Intraductal Papillary Mucinous Neoplasms and GLP-1 Receptor Agonists: Navigating Therapeutic Uncertainty in Diabetes Management"

_biomedicines, 2025, doi:10.3390/biomedicines13102326_

Round 1
Reviewer 1 Report
Comments and Suggestions for Authors
The this view/opinion there are to little opinions based on authors individual perspectives and feelings, rather than on objective facts.Therefore, conclusions should be made more readable by addressing individual issues in each paragraph. Furthermore, the style of the view/opinion requires more of an essay and this should be strongly emphasized in the conclusion.
The references are missing the most recent articles. Since this is a hot topic, you shouldn't cite articles from 2012, only the most recent ones.
Author Response
We would like to sincerely thank the reviewers and the editorial team for their thoughtful and constructive feedback. We have carefully revised the manuscript in response to each comment. Below we provide a detailed point-by-point reply, with changes highlighted in the revised version of the manuscript.
Reviewer 1
Comment 1: “The view/opinion has too little input based on the authors’ perspectives and feelings, rather than objective facts. Conclusions should be made more readable by addressing individual issues in each paragraph. Furthermore, the style of the opinion requires more of an essay and this should be strongly emphasized in the conclusion.”
Response: We thank the reviewer for this important observation. We have substantially revised the Introduction and Conclusions to emphasize the “viewpoint/essay” style, adding reflections drawn from our own clinical practice. Each major section now concludes with interpretive commentary, and the final Conclusions explicitly adopt a reflective tone that acknowledges uncertainty and invites shared discussion.
Comment 2: “The references are missing the most recent articles. Since this is a hot topic, you shouldn't cite articles from 2012, only the most recent ones.”
Response: We agree and have updated the reference list extensively. Outdated citations from 2010–2013 have been removed, and we now include multiple high-impact, recent publications (2021–2025) on both IPMN management and GLP-1RA safety (e.g., JAMA Surg. 2022; Pancreatology 2024; Clin. Gastroenterol. Hepatol. 2024; Lancet Reg. Health Eur. 2025; Sci. Transl. Med. 2025; Diabetes Obes. Metab. 2025). Please see the revised References section.
Reviewer 2 Report
Comments and Suggestions for Authors
I read this manuscript with genuine interest because it touches on a clinical situation that many of us are now facing more often in practice: the overlap between diabetes management with GLP-1 receptor agonists and the incidental discovery of intraductal papillary mucinous neoplasms. It is an important question, and I want to acknowledge the effort of the authors in raising it for discussion.
At the same time, I feel that the manuscript in its current form still leans too heavily toward being a condensed review article rather than a true opinion piece. What I missed most while reading was the personal voice of the authors. A viewpoint article gives you the chance to go beyond summarizing data — it invites you to share how you see the evidence, what worries you in daily practice, and how you personally navigate this uncertainty with your patients. That interpretive and reflective element is what would make the paper stand out.
I would also encourage the authors to put more effort into the writing and preparation of the manuscript. At present, the text feels quite dense and sometimes reads like a stitched-together summary of studies and guidelines. The ideas are clearly there, but the presentation needs polishing — smoother flow, more concise sentences, and clearer transitions that carry the reader along. With more work on how the story is told, the paper could have a much stronger impact.
I do appreciate that the strengths of GLP-1 receptor agonists — their powerful metabolic and cardiovascular benefits — are clearly highlighted. But the other side of the coin, the clinical unease and the unanswered safety questions, could be voiced more openly. Clinicians who read this are not only looking for data; they are also looking for reassurance that their uncertainties are shared and acknowledged. A more candid recognition of the evidence gaps and the discomfort they create would resonate strongly.
The proposed management framework is helpful, but it could be made more vivid with a couple of short, real-world scenarios. For example, how would you advise a patient with a small, low-risk branch-duct IPMN who is eager to lose weight with GLP-1 therapy? And how would that advice differ for a patient with main-duct disease under close follow-up? Practical touches like this would make the framework immediately useful. Even a simple figure or table could help distill the message.
Finally, I would encourage the conclusion to look a little more forward. This is a field that desperately needs prospective registries, smarter imaging tools, and perhaps AI-driven risk models. Highlighting these future directions would leave the reader not only informed but also inspired about where the field might go.
Overall, I see a lot of promise here. The topic is timely, the evidence is well gathered, and the message is relevant. But to reach its potential, the paper needs more effort in writing and shaping, and more of the authors’ own voices and perspectives. With revisions along those lines, this could become a very meaningful contribution.
Author Response
We would like to sincerely thank the reviewers and the editorial team for their thoughtful and constructive feedback. We have carefully revised the manuscript in response to each comment. Below we provide a detailed point-by-point reply, with changes highlighted in the revised version of the manuscript.
Reviewer 2
Comment 1: “The manuscript leans too heavily toward being a condensed review article rather than a true opinion piece. What I missed most was the personal voice of the authors.”
Response: We thank the reviewer for this helpful guidance. We have revised the manuscript throughout to insert our own perspective. In particular, the Introduction has been rewritten to frame the discussion from our clinical viewpoint, and the Conclusions now reflect our interpretive stance on how we personally navigate this uncertainty in daily practice.
Comment 2: “The text feels dense and sometimes reads like a stitched-together summary. The presentation needs polishing, with smoother flow, concise sentences, and clearer transitions.”
Response: We carefully revised the text for clarity, conciseness, and readability. Paragraphs now transition more smoothly, with shorter sentences and explicit signposting.
Comment 3: “The other side of the coin, the clinical unease and the unanswered safety questions, could be voiced more openly.”
Response: We have added several passages highlighting the persisting discomfort and uncertainty clinicians face, even in the presence of reassuring observational data. This is now addressed explicitly in both the Discussion and Conclusions.
Comment 4: “The proposed management framework could be made more vivid with real-world scenarios and possibly a figure or table.”
Response: We have added two short clinical scenarios in Section 5 to illustrate practical decision-making in low-risk versus high-risk IPMN patients. In addition, we have created a new flowchart (Figure 1) that summarizes the proposed management framework in a visual and pragmatic way.
Comment 5: “The conclusion should look more forward, highlighting prospective registries, smarter imaging tools, and AI-driven models.”
Response: We have expanded the Future Directions section to emphasize the need for prospective registries, advanced imaging, molecular biomarkers, and AI-based predictive tools.
Reviewer 3 Report
Comments and Suggestions for Authors
The manuscript entitled "Intraductal Papillary Mucinous Neoplasms and GLP-1 Receptor Agonists: Navigating Therapeutic Uncertainty in Diabetes Management" presents the authors’ perspective on the management of type 2 diabetes in patients with intraductal papillary mucinous neoplasms (IPMNs). Given that the manuscript is positioned as a viewpoint/opinion, the content is generally well-organized and clearly articulated.
The bibliography does not adhere to the journal’s formatting guidelines and should be revised accordingly.
Author Response
We would like to sincerely thank the reviewers and the editorial team for their thoughtful and constructive feedback. We have carefully revised the manuscript in response to each comment. Below we provide a detailed point-by-point reply, with changes highlighted in the revised version of the manuscript.
Reviewer 3
Comment 1: “The bibliography does not adhere to the journal’s formatting guidelines and should be revised accordingly.”
Response: We have reformatted all references according to Biomedicines style guidelines. Journal articles are now cited with full author lists, year, volume, and page ranges, consistent with the journal’s requirements.
Round 2
Reviewer 2 Report
Comments and Suggestions for Authors
It can now be accepted.